# A Lightweight Recognition Method for Rice Growth Period Based on Improved YOLOv5s

**DOI:** 10.3390/s23156738

**Published:** 2023-07-27

**Authors:** Kaixuan Liu, Jie Wang, Kai Zhang, Minhui Chen, Haonan Zhao, Juan Liao

**Affiliations:** 1College of Engineering, Anhui Agricultural University, Hefei 230036, China; 22720252@stu.ahau.edu.cn (K.L.); zhangk@stu.ahau.edu.cn (K.Z.); cmh8586@163.com (M.C.); 22721335@stu.ahau.edu.cn (H.Z.); 2Anhui Provincial Rural Comprehensive Economic Information Center, Hefei 230031, China; jiewong629@sina.cn; 3Hefei Institute of Technology Innovation Engineering, Chinese Academy of Sciences, Hefei 230094, China

**Keywords:** growth period recognition, rice, deep learning, YOLOv5s, lightweight model

## Abstract

The identification of the growth and development period of rice is of great significance to achieve high-yield and high-quality rice. However, the acquisition of rice growth period information mainly relies on manual observation, which has problems such as low efficiency and strong subjectivity. In order to solve these problems, a lightweight recognition method is proposed to automatically identify the growth period of rice: Small-YOLOv5, which is based on improved YOLOv5s. Firstly, the new backbone feature extraction network MobileNetV3 was used to replace the YOLOv5s backbone network to reduce the model size and the number of model parameters, thus improving the detection speed of the model. Secondly, in the feature fusion stage of YOLOv5s, we introduced a more lightweight convolution method, GsConv, to replace the standard convolution. The computational cost of GsConv is about 60–70% of the standard convolution, but its contribution to the model learning ability is no less than that of the standard convolution. Based on GsConv, we built a lightweight neck network to reduce the complexity of the network model while maintaining accuracy. To verify the performance of Small-YOLOv5s, we tested it on a self-built dataset of rice growth period. The results show that compared with YOLOv5s (5.0) on the self-built dataset, the number of the model parameter was reduced by 82.4%, GFLOPS decreased by 85.9%, and the volume reduced by 86.0%. The *mAP* (0.5) value of the improved model was 98.7%, only 0.8% lower than that of the original YOLOv5s model. Compared with the mainstream lightweight model YOLOV5s- MobileNetV3-Small, the number of the model parameter was decreased by 10.0%, the volume reduced by 9.6%, and the *mAP* (0.5:0.95) improved by 5.0%—reaching 94.7%—and the recall rate improved by 1.5%—reaching 98.9%. Based on experimental comparisons, the effectiveness and superiority of the model have been verified.

## 1. Introduction

Rice is a major food crop for the people of the world and is crucial for food security and social stability, and the improvement of rice yield and quality has always been focused on strategic agricultural development [1]. One method for the improvement of rice yield and quality is based on real-time and accurate rice growth period information and its environment condition during extraction [2]. As one of the important data in terms of rice growth, the recognition of rice growth period information refers to the record of morphological changes in rice during different growth and development stages, which is of great significance for cultivators to properly time field operations, including fertilizer, irrigation, weed control, and other agricultural activities to achieve high-yield and high-quality rice. Additionally, the rice growth period information can help people analyze the relationship between rice growth processes in the field and agro-meteorological conditions, thereby providing efficient agricultural services and improving rice yields [3]. However, the traditional manual field observation of the rice growth period is mainly achieved by human inspection, using visual assessments or basic measurements, which is time-consuming, labor-intensive, and prone to large human errors, especially in a long-term observation process. Therefore, it is necessary to study automatic observation methods of rice growth period to effectively reduce the information acquisition cost, improving accuracy and field management efficiency.

With the development of remote sensing technology, remote sensing data based on satellites and aerial imagery have been widely used in crop growth monitoring such as rice [4,5], wheat [6,7], and maize [8,9]. However, remote sensing technology is suitable for obtaining a large area of ground information to recognize crop growth period with reference to the overall situation of crop growth and is not suitable for ground-based observation of the crop growth stage because of its characteristics of long-distance imaging and limited spatial resolution. Moreover, the continuous acquisition of multi-temporal remote sensing data increases the difficulty of the implementation of the scheme in agriculture. Compared with remote sensing technology, computer vision technology can obtain high-resolution images and observe the growth status of crops in a small area [10], where the crop growth period can be determined by observing changes in the phenotypic characteristics of the crop. Research has shown that key crop growth period monitoring can be estimated from these images acquired from digital high-resolution cameras by evaluating the structure and texture of the plants in detail [11]. Bai et al. [12] used a support vector machine and diffusion-convolution neural networks to distinguish image patches of rice ears, where the number of spikes detected determined the heading data of the rice. Zhang et al. [13] proposed an automatic rice tiller period recognition algorithm combining principal component analysis and a support vector machine. Keller et al. [14] built an automatic method of estimating soybean leaf coverage from images by assessing a number of different machine-learning- and threshold-based algorithms. But all of the above methods require prior knowledge to select recognition features manually and extract images with low-level features. Then, classification can be used to obtain classification results, in which their success was dependent on prior knowledge of botany and crucial parameters, and thereby, they were of limited value for real applications in the recognition of crop growth period.

By virtue of innovations in deep neural computing algorithms, CNNs can learn deeper robust features automatically from the original image, which has achieved impressive results in the image recognition of agriculture [15]. A study documented in [16] investigated a wheat and barley growth stage estimation by the classification of proximal images using convolutional neural networks (ConvNets), which indicated that the classification accuracy of the transfer learning ConvNet model was 99.7%. A method for recognizing the cotton growth period was proposed based on an optimized VGG model in [17], whose accuracy was 93.27%. In the aforementioned methods, the researchers used CNNs to automatically recognize the crop growth period instead of using human inspection and traditional image processing methods. While the above methods have performed well in the recognition of crop growth period, they focus solely on either crop growth period image identification or classification, and few works have considered the problem of the high requirements for the computing power of these devices.

Due to the real-time requirements and limited computing power in practical agricultural equipment, a model with a large number of model parameters and slow detection speed is not suitable for practical agricultural scenarios. In the background of modern intelligent agriculture, to reduce labor costs and improve agricultural production efficiency and precision, the important factor that affects the promotion of automation and intelligence is the cost of equipment hardware. In addition, the canopy density and morphology of rice in paddy fields will change with time, and the transition between adjacent growth periods is difficult to accurately classify because of the similarities in adjacent growth periods. Herein, it is necessary to build a model for rice growth period recognition with a balance between accuracy, real-time operation, and low hardware cost requirements. In recent years, You Only Look Once (YOLO) series networks have been proposed and widely used in agriculture due to their ability to detect efficiently and accurately. In [18], an improved YOLOv4 method for identifying the growth period of strawberries in a greenhouse environment was presented, and the results showed that the detection accuracy (*AP*) of the model during the flowering, fruit expansion, green, and mature period was 92.38%, 82.45%, 68.01%, and 92.31%, respectively. Tian et al. [19] designed a system based on YOLOv3 that can detect apples at three different stages in the orchard in real time. Roy et al. [20] designed a high-performance, real-time, fine-grained target detection framework that can address obstacles such as dense distribution and irregular morphology, which is based on an improvement of YOLOv4. Ahmed [21] used YOLOv5, YOLOR, and Faster R-CNN deep learning network models in detecting pavement defects, where the resulting analysis showed that the YOLOv5 model was highly reliable and stable. To the best of our knowledge, the YOLO family has already been widely used in the detection and classification of agricultural crops, which motivates us to consider YOLOv5 as a baseline model. However, there are scant studies that apply the YOLO family to the recognition of the rice growth period.

The key objective of this work is to develop an effective and lightweight scheme for differentiating the five growth stages of rice in the growing period using deep learning techniques. For field rice object recognition, the DarkNet structure in the YOLO model will produce many redundant feature maps, which have a limited impact on detection and increase the model’s parameters, making the YOLO model run very slowly on some edge devices [22]. To solve the real-life problems of identifying rice growth periods, this study proposes a lightweight improved model based on YOLOv5s: Small-YOLOv5, which achieves excellent results in being lightweight, its accuracy, and it has the significance of further practical promotion.

## 2. Materials and Methods

### 2.1. Image Collection 

The rice growth period images were obtained from 12 experimental fields of the Hefei Branch of the Anhui Agricultural Meteorological Center, China [23], and the size of each field was 12 m × 5 m, as shown in Figure 1a. In 2019 and 2021, according to the requirements of the rice sowing period, four different single-crop rice varieties (Danjing10, Xuanjingnuo1, Chuangliangyou699, and Liangyou631) were planted in the 12 fields. Complete process images were collected from transplantation to harvest using network cameras, which were set up at the diagonal points of the rice field, as shown in Figure 1b. The camera used was the Hikvision iDS-2DF8825IX-A(T5). The camera video output supported 3840 × 2160 @25FPS, 2100 lines, 37× optical zoom, a maximum of 300 preset positions, 18 cruise paths, and a mounting riser 2.5 m above ground [23]. The images acquired from the camera were saved in RGB color space in JPG format. Ten shooting preset points were deployed in the experimental area. The camera timely captured the fixed point and uploaded it to the FTP server to minimize substantial changes in light intensity caused by direct sunlight [24]. 

### 2.2. Construction of Rice Growth Period Dataset

The camera mentioned above was used to track captured images of the growth period of rice after transplantation in the paddy field in 2019 and 2021. The images acquired from the camera were saved in RGB color space in JPG format, which contained seven rice growth periods, including the turning green period, tillering period, jointing period, booting stage, heading period, milky period, and maturity period. Among these growth periods, the turning green period, tiller period, jointing period, heading period, and milky period are the key stages in the growth and development of rice. Hence, in this study, the proposed model was built to identify these five growth periods. Figure 2 gives samples of these five growth periods. The resolution of the original images was 3840 × 2160 pixels, which was too large for processing, especially for the deep-learning-based method, so it was necessary to shrink the size of images to reduce the computational load. Then, a total of 4844 images with a resolution of 640 × 640 pixels were cut out from the original images of the five growth periods. To train the model with the training dataset and calculate the performance with the validation dataset, the training dataset and validation dataset were randomly allocated in a ratio of 8:2 for each growth period, where the number of images for each set of each growth period is shown in Table 1.

### 2.3. Improved YOLOv5s Network Structure

YOLOv5 is the most accurate model in the YOLO series of networks in recent years [25]. Its praised features include its more systematic and less redundant code. YOLOv5 has been widely and maturely used in the agricultural field, especially for crop phenotype research. There are four versions of YOLOv5: YOLOv5s, YOLOv5m, YOLOv5l, and YOLOv5x. Among them, YOLOv5s is the fastest and smallest, and it retains a high degree of accuracy. It has a strong comprehensive performance. Therefore, to achieve high precision, low parameters, and fast detection for rice growth period recognition, this paper chooses YOLOv5s for optimization and improvement.

One of the objectives of improving the rice growth period recognition model is to reduce the number of parameters and GFLOPs by lightweighting, and the other objective is to improve the detection speed to achieve these two objectives while still maintaining excellent detection accuracy. The backbone network in YOLOv5 consists of CBS and CSP modules [26], which can effectively extract features from images but introduce a large number of model parameters, thus increasing the computational load; some of these model parameters are redundant for feature extraction of rice images in the paddy field, therefore, it is necessary to eliminate this redundant feature map to reduce the size of the rice growth period recognition. In the view of the characteristics of rice images, by comparing different convolutional structures, MobileNetV3 [27] is first selected as the backbone network of the model. The improved backbone network effectively reduces the number of parameters in the model, thus reducing the computational complexity of the model to achieve the purpose of being lightweight, and at the same time, greatly improves the detection speed. Second, although the number of model parameters in the backbone network has been reduced, there are still many model parameters in the feature fusion process of the neck network, which affects the speed of the feature fusion. Considering the characteristics of the feature fusion, a new method was designed to replace the neck network, i.e., the standard Conv module was replaced by the GsConv module [28], which improves the speed of the feature fusion, reduces the computational load of the model, achieves further lightweighting of the model, and improves the detection speed of the model while improving indicators such as *mAP* (0.5:0.95) and recall significantly. The structure of the lightweight recognition model for the rice growth period based on YOLOv5s proposed in this study is shown in Figure 3.

#### 2.3.1. Improvements of Backbone Networks

In 2019, Howard et al. [27] proposed MobileNetV3 networks. As the latest version of the MobileNet series, it has small parameters, high accuracy, and fast real-time detection speed, and it is widely used in embedded and mobile terminals. MobileNetV3 is composed of multiple basic Block modules stacked together, and v3-Large and v3-Small structures are formed by stacking different numbers and parameterized Block modules. There are a total of 15 Block modules in the MobileNetV3-Large model, while there are 11 in MobileNetV3-Small, with fewer channels. The MobileNetV3-Small structure is shown in Figure 4. Based on the actual rice field recognition scenario, the MobileNetV3-Small structure was used as the backbone of the improved YOLOv5s in this paper. The feature map in MobileNetV3 will be scaled 5 times, so the feature map output by the last convolutional layer is only 1/5 of the size of the input image. 

In the MobileNetV3-Small model, the 11 Blocks have two different structures, as shown in Figure 5. From Figure 5a, it can be found that the feature map first goes through a 1 × 1 convolution, which compresses the number of channels in the feature map. Then, a depthwise separable convolution is applied to the feature map to reduce the computational cost, where n is 3 or 5. In traditional convolution, each convolutional kernel will convolve each channel of the feature map once, while in depthwise separable convolution, each convolutional kernel will convolve only a single channel of the feature map, greatly reducing the computational cost. After the depthwise separable convolution operation, the feature map is connected to a 1 × 1 convolution, which specifies the number of channels and aggregates the feature map. Finally, the output feature maps of the two paths are added together to obtain the final feature map. The structure shown in Figure 5a is used in second and third Block of the MobileNetV3-Small module, and the remaining Block uses the Structure2 as shown in Figure 5b. Comparing Figure 5a with Figure 5b, the SE module [29] is introduced in Figure 5b, which is to enhance the semantic information of the target area. 

Depthwise separable convolution (DSC) [30]. An important part of MobileNetV3 is the DSC, and its core idea is to decompose the standard convolution operation into depthwise convolution and pointwise convolution, as shown in Figure 6. DSC can be divided into two parts: depthwise convolution and pointwise convolution, as shown in the Figure 6. The depthwise convolution is completely performed on a two-dimensional plane, and each channel is convolved by only one convolution kernel, where the number of convolution kernels is equal to the number of channels in the previous layer, which cannot effectively utilize the feature information of different channels at the same spatial position. The pointwise convolution operation is very similar to traditional convolution operation, where the convolution operation can weigh and combine the feature map in the depth direction of the previous step, generating a new feature map [30]. For several convolution kernels, there are several output characteristic maps. The depthwise convolution uses different convolution kernels for each input channel, that is, the number of groups in the network is equal to the number of channels in the network, thereby reducing the computational complexity of the convolution [31]. Assuming the size of the convolution kernel is $D_{k}\times D_{k}$, *M* is the number of input channels, *N* is the number of output channels, and $D_{F}\times D_{F}$ is the size of the output feature map, the computation of standard convolution and DSC are calculated in Formulas (1) and (2), respectively.
(1)$$D_{k}\times D_{k}\times M\times N\times D_{F}\times D_{F}$$
(2)$$D_{k}\times D_{k}\times M\times D_{F}\times D_{F}+D_{F}\times D_{F}$$

Compared with ordinary convolution, the computation amount of DSC is reduced as shown in Formula (3). From Formula (3), it can be seen that the computation reduces greatly by using DSC.
(3)$$\frac{D_{k}\times D_{k}\times M\times D_{F}\times D_{F}+D_{F}\times D_{F}}{D_{k}\times D_{k}\times M\times N\times D_{F}\times D_{F}}=\frac{1}{N}+\frac{1}{D_{k}^{2}}$$

h-swish activation function and Squeeze-and-Excitation (SE) module. h-swish is an improvement on the swish function [32], where the sigmoid function is replaced by ReLU6(*x* + 3)/6. This replacement reduces problems such as gradient disappearance caused by the increase in network layers. It can also greatly reduce computational complexity, improve model performance, and increase model detection efficiency. The core idea of the SE attention mechanism [29] is to model the interdependence between channels and generate corresponding weights for each channel to improve significant features and suppress unimportant features. The SE attention mechanism network consists of two steps: Squeeze and Excitation. Squeeze compresses the current feature map into a global compression feature vector by performing global average pooling (GAP) on the extracted features. Excitation obtains the normalized weights of each channel through two fully connected layers and multiplies the weighted features as inputs to the next layer of the network. The input *X* has a size of *H* × *W* × *C*, where *C* is the number of feature channels, and *H* × *W* is the height and width of the feature map. Fc represents a fully connected layer, ReLU and h-swish are activation functions, and *Y* multiplies the weight coefficients generated for each channel with all elements of the corresponding channel. It enhances important features, weakens unimportant features, and makes the extracted features more directional. The SE attention mechanism structure is shown in Figure 7a.Inverted residual structure with linear bottlenecks. It first uses a 1 × 1 convolution to reduce dimensionality, then extracts features through a 3 × 3 convolution, and finally uses a 1 × 1 convolution to increase dimensionality in the residual structure. This results in a structure that resembles an hourglass with a small middle and two large ends [33]. However, in the inverted residual structure, dimensionality is increased first using a 1 × 1 convolution, then features are extracted using a 3 × 3 DSC, and finally dimensionality is reduced using a 1 × 1 convolution. The order of dimensionality reduction and increase is swapped, and SC is replaced by DSC convolution, resulting in a shuttle-shaped structure with a small middle and two large ends. In addition, after dimensionality reduction in the convolution layer, non-linear transformations, such as ReLU, are not added in order to avoid information loss as much as possible. The purpose of this is to minimize the risk of losing information. The inverted residual structure is shown in Figure 7b.

#### 2.3.2. Improvement of Neck Network

Many lightweight optimization designs reduce parameter and FLOP count through depthwise separable convolution (DSC), which is effective but has obvious disadvantages: the channel information of the input image is separated during calculation, which is amplified in the direct network backbone. To make DSC as close to the standard convolution (SC) as possible, Li et al. [28] proposed GsConv. As shown in Figure 8, Shuffle is used to permeate the information generated by SC (dense convolution operation) [34] into every part of the information generated by DSC. This allows the information from SC to be completely mixed into the output of DSC.

To speed up the prediction calculation speed, the feed images in CNN must undergo a similar transformation process in the backbone: Spatial information is gradually transmitted to the channels. Each time, the spatial (width and height) compression and channel expansion of the feature map will lead to the loss of some semantic information. Dense convolution calculation maximally preserves the hidden connections between each channel, while sparse convolution completely severs these connections. GsConv tries to preserve these connections as much as possible. However, if it is used in all stages of the model, the network layers of the model will be deeper, and deep layers will increase the resistance to data flow, significantly increasing the inference time. When these feature maps reach the neck, they have become long and thin (with the maximum channel dimension and minimum width and height dimensions) and no longer need to be transformed. Using GsConv to process the concatenated feature maps has the benefits of less redundant information and fewer compressions, resulting in better feature fusion effect.

## 3. Experimental and Analysis

### 3.1. Experimental Setup

The environment for this experiment was the Windows 11 operating system. The model algorithm was implemented using the PyTorch deep learning framework, with a torch version of 1.13 and a CUDA version of 11.6. The graphics card used was NVIDIA GeForce RTX 3090, the CPU was Intel I7-11700K, and the memory was 32GB DDR4 3200. During the training process, the input image was set to 640 × 640, and the SGD was used as the optimization function to train the model. The model training epoch was 200, with a batch size of 128, and an initial learning rate of 0.02. The same data augmentation algorithm as the original YOLOv5 algorithm was used in this experiment.

### 3.2. Evaluation Metrics

The evaluation metrics used in this study include precision (*P*), recall (*R*), mean average precision (*mAP*), number of parameters (Params), and floating point operations per second (GFLOPs). Among them, precision and recall are the basic metrics, and the calculated *mAP* based on precision and recall is used as the evaluation metric to measure the model’s recognition accuracy. GFLOPs are used to measure the complexity of the model or algorithm, and Params represent the size of the model. Generally, the smaller the Params and GFLOPs, the less computing power the model requires, and the lower the hardware requirements, the easier it is to build on low-end devices.
(4)$$P=\frac{T_{P}}{T_{P}+F_{P}}$$
(5)$$R=\frac{T_{P}}{T_{P}+F_{N}}$$

In Formulas (4) and (5), *T_P_* means a correct detection result: predicted bounding boxes are present near the labeled bounding box, and the IoU (intersection-over-union) between the predicted and labeled bounding boxes is greater than the crossover ratio threshold. *F_P_* means an incorrect detection result: predicted bounding boxes are present near the marker bounding box, but the *IoU* between the predicted and labeled bounding box is less than the intersection threshold. *F_N_* means the detection network does not output a prediction near a labeled bounding box.
(6)$$AP=\frac{1}{m}\sum_{i}^{m}P_{i}=\frac{1}{m}\times P_{1}+\frac{1}{m}\times P_{2}+\cdots +\frac{1}{m}\times P_{m}$$

Under Formula (6), assuming there are m positive events among these *N* samples, we obtain the m recall values (1/*m*, 2/*m*, …, 1). For each recall value *R*, we can calculate the corresponding maximum precision value *P*, then average these *m* precision values to obtain the final *AP* value. *mAP* is calculated by averaging the *AP* values of all classes in the dataset, as shown in Formula (7).
(7)$$mAP=\frac{1}{C}\sum_{C}^{j}P_{j}$$

The object detection task also includes a bounding box regression task, which is usually tested for its accuracy. The calculation method of *IoU* is shown in Formula (8)
(8)$$IoU=\frac{S_{1}}{S_{2}}$$
where *S*_1_ is the overlap area between the predicted box and the ground-truth box, and *S_2_* is the total area occupied by the predicted boxes and the ground-truth box. *mAP* (0.5) is the mean average precision when the *IoU* threshold is set to 0.5, while *mAP* (0.5:0.95) represents the *mAP* at different *IoU* thresholds ranging from 0.5 to 0.95 with a step size of 0.05.

### 3.3. Self-Comparison on Improved YOLOv5s

#### 3.3.1. Influence on MobileNetV3 and GsConv

We investigated the performance of different modules in the improved YOLOv5s using the YOLOv5s as the backbone network design on the self-built dataset. Table 2 shows the number of model parameters, GFLOPs, precision, recall, and mean average precision indicators obtained with different models. From the table, it is noted that compared with YOLOv5s, using only the improved backbone network MobileNetV3 reduces the number of parameters by 80.3% and GFLOPs by 84.7%, while precision, recall, *mAP* (0.5), and *mAP* (0.5:0.95) all decrease, except for an 8.0% decrease in *mAP* (0.5:0.95). Moreover, the model using only the improved neck network GsConv reduces the number of parameters by 6.8% and GFLOPs by 6.8% compared to YOLOv5s, but performance metrics in terms of precision, recall, *mAP* (0.5), and *mAP* (0.5:0.95) decrease significantly, much greater than when using MobileNetV3 alone. For the design of the combination of YOLOv5s, MobileNetV3, and GsConv in this study, the model achieves the smallest value in terms of the number of model parameters and GFLOPs, which are reduced by 82.4% and 85.9%, respectively, compared with YOLOv5s. In terms of accuracy, *mAP* (0.5) is the same as YOLOv5s, only a slight decrease in recall and *mAP* (0.5:0.95). According to the above analysis, the design combination of MobileNetV3 and GsConv in the YOLOv5s is more lightweight than other three models.

#### 3.3.2. Recognition Results of five Growth Periods of Rice

To get better intuition of the recognition results of the five growth periods of rice using the proposed model, several images were randomly selected from the acquired videos that were different from the images in the dataset for detailed comparison, and the visual recognition results using the proposed model are presented in Figure 9. It was observed that the proposed model could accurately recognize different growth periods of rice, where the recognition accuracy was higher than 95%. Combined with Table 2, the proposed model reduced the number of model parameters while maintaining model accuracy. This is because the proposed model introduced MobileNetV3 as a backbone network and used GsConv to replace the neck network of the YOLOv5s, which can lightweight the model and better fuse image features. In addition, Figure 10 gives the P–R curves of different grow periods of rice, where the values of the horizontal and vertical axes are the recall and accuracy calculated using Formulas (4) and (5). As shown in Figure 10, it was noted that the accuracy basically does not decrease when the recall rate is in the 0–90% range. And for different growth periods, the area difference between the P–R curves and the coordinate axis circumference of the model in this paper is very small, which indicates that the proposed model can achieve a balance between accuracy, real-time operation, and low hardware cost requirements.

### 3.4. Comparison of Different Models

In order to better verify the overall performance of the proposed model (named Small-YOLOv5), we compared the recognition results of the proposed model with four other models, Faster R-CNN, YOLOv4, YOLOv7 and YOLOv8. In the comparison experiment, the same dataset and loss function mentioned above were used for the four models, and the evaluation indexes including the number of parameters, GFLOPs, precision, recall, *mAP* (0.5), and *mAP* (0.5:0.95) were applied to examine the four different models, as shown in Table 3. From the table, it was noted that the number of parameters and GFLOPs of the Small-YOLOv5s were menial compared with other models, which proves that the design of using depth separable convolution and GsConv in Small-YOLOv5s is more lightweight. As a classic two-stage convolutional neural network, Faster R-CNN proposed the region selection network to generate candidate boxes, which could improve the generation speed of detection boxes, but false detection of candidate boxes in the first stage could lead to decreased detection performance of the model. Moreover, Faster R-CNN still had some computational redundancy in the detection stage compared with Small-YOLOv5s. In the YOLOv4, CSPNet (Cross Stage Partial Networks) was used as the backbone network to solve the optimization gradient information repetition problem from other large convolutional neural network frameworks and backbone networks, thus reducing the number of parameters and value of GFLOPs, but it is still not as lightweight as the Small-YOLOv5. In the YOLOv7, Extended-ELAN (extended efficient layer aggregation network) was constructed to improve the detection accuracy of the model, which was slightly higher than that of Small-YOLOv5 but leads to a larger number of parameters and GFLOPs that are hard to widely use in agriculture. For the YOLOv8, it is a more complex network architecture, including multiple residual units (Residual Unit) and multiple branches, which has higher performance in accuracy, but its model is also more complex with higher Params and GFLOPs, as well as higher requirements for agricultural hardware equipment.

Moreover, in order to verify the overall performance of the proposed model, we compared it with four mainstream lightweight object detection models, YOLOv4-tiny, YOLOv7-tiny, YOLOv5n, and YOLOv8n. In the comparison experiment, the same dataset was used for four detection models, and the evaluation indexes introduced in Section 3.2 were applied to examine the models. The performances of the different models are shown in Table 4, where the values of the proposed model in this study are highlighted in bold. From the experimental results, it can be seen that the performance of the proposed model Small-YOLOv5 is significantly better than that of the other four models. While YOLOv8n achieved excellent accuracy, there is still a big gap between the lightweight and Small-YOLOv5; the YOLOv7-tiny performs the worst in terms of the lightweight evaluation measure, and YOLOv4-tiny performs the worst in terms of the accuracy evaluation measures. Comparatively, the number of model parameters and GFLOPs of the Small-YOLOv5 are 1.24 MB and 2.3, which are lower than YOLOv5n by 29.6% and 43.9%, respectively. And precision, recall, and *mAP* (0.5:0.95) of the Small-YOLOv5 are 96.2%, 98.9%, and 96.6%, which are higher than YOLOv5n by approximately 1.7%, 4.0%, and 2.7%, respectively—only slightly lower than YOLOv5n in *mAP* (0.5). Hence, the effectiveness of the Small-YOLOv5s in lightweighting was demonstrated without affecting the detection accuracy.

## 4. Conclusions

To address the challenge of detection models in rice growth period recognition, an improved YOLOv5s, based on the basic YOLOv5s, was built to achieve a lightweight identification model for rice growth period in this study. Firstly, we collected rice plant images in 12 experimental fields with self-installed network cameras and built a dataset of five key growth periods of rice. To reduce the number of model parameters while maintaining model accuracy, we then designed the backbone network with the MobileNetV3 module and introduced the GsConv module with lower computation intensity in a YOLOv5s neck network to accelerate feature fusion and reduce computational complexity. Experiments delivered on the rice growth period image dataset have shown that compared with the traditional YOLOv5s, the proposed model can effectively reduce the complexity of the deep learning detection model and not affect the recognition accuracy. The precision, recall, *mAP* (0.5), as well as *mAP* (0.5:0.95) index were chosen to evaluate the model recognition performance, whose scores were 96.2%, 98.9%, 98.7%, and 94.2%, respectively. Furthermore, compared with YOLOv4, YOLOv7, Faster R-CNN, YOLOv4-tiny, YOLOv7-tiny, and YOLOv5n, the proposed model is a lightweight model for different growth periods of rice, which provides support toward the rice field management, rice yield improvement, breeding and cultivation, etc., and it is suitable for intelligent agricultural equipment with limited hardware system resources.

The recognition accuracy of the proposed model needs to be further improved in a complex paddy field environment. Improving the dataset and model optimization will be the focus of our next research. For the dataset, an automatic data enhancement method can be introduced to improve the diversity of dataset. And in terms of the model optimization, we plan to design a lightweight attention mechanism to make the model pay more attention to the recognition target and to reduce the extraction of background information, which can improve the precision of the model.

## Figures and Tables

**Figure 1 sensors-23-06738-f001:**
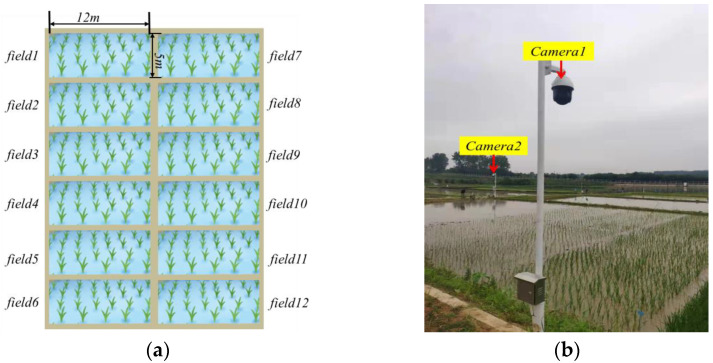
Image acquisition. (**a**) Distribution diagram of 12 experimental fields; (**b**) image acquisition device.

**Figure 2 sensors-23-06738-f002:**
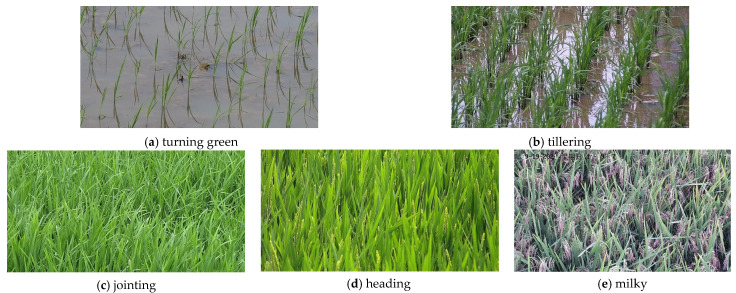
Sample images of rice at different growth periods.

**Figure 3 sensors-23-06738-f003:**
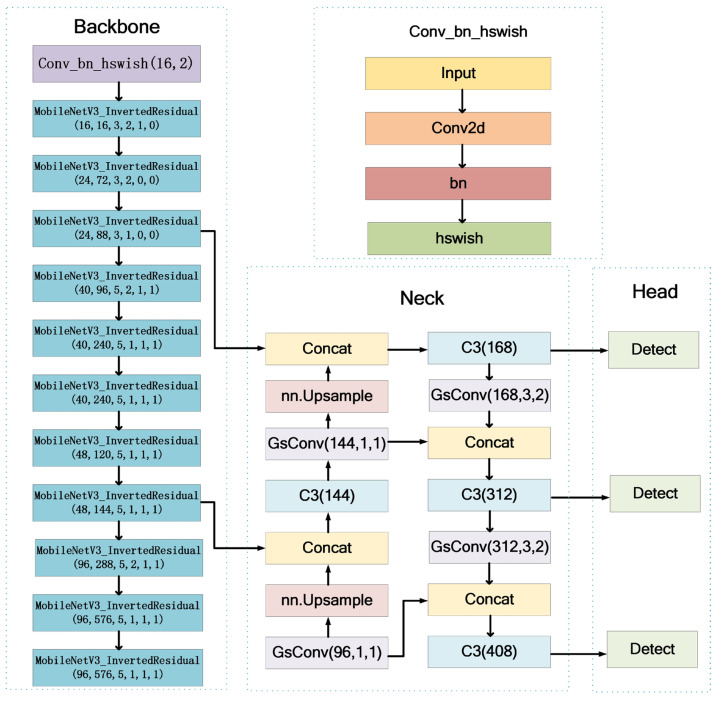
The structure of the improved YOLOv5s.

**Figure 4 sensors-23-06738-f004:**
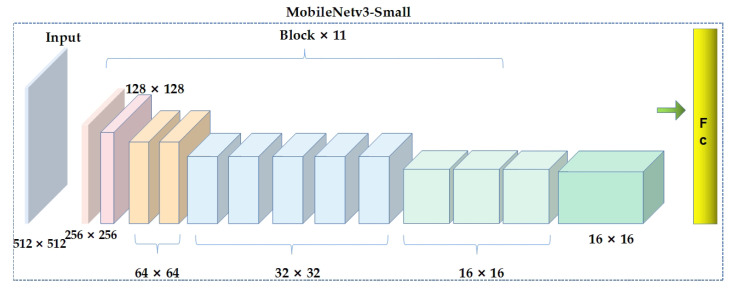
Structure of MobileNetV3-Small module.

**Figure 5 sensors-23-06738-f005:**
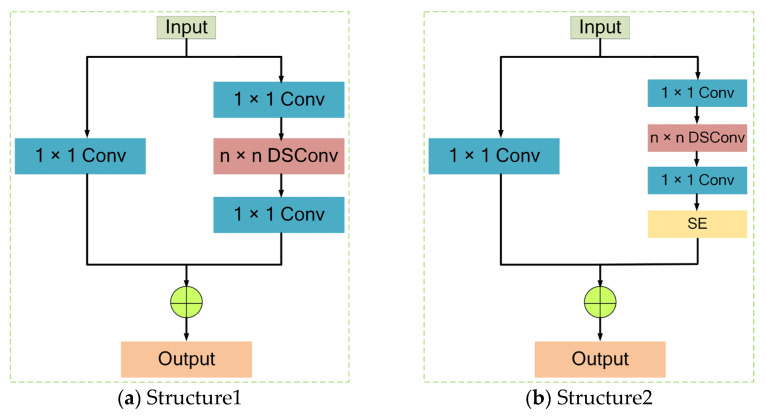
Two different structures in the MobileNetV3 module.

**Figure 6 sensors-23-06738-f006:**
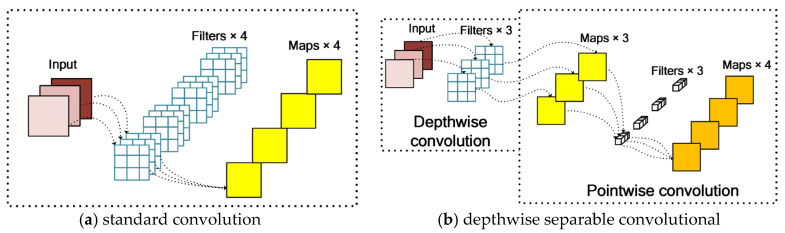
Structures of a standard convolution and depthwise separable convolutional block.

**Figure 7 sensors-23-06738-f007:**
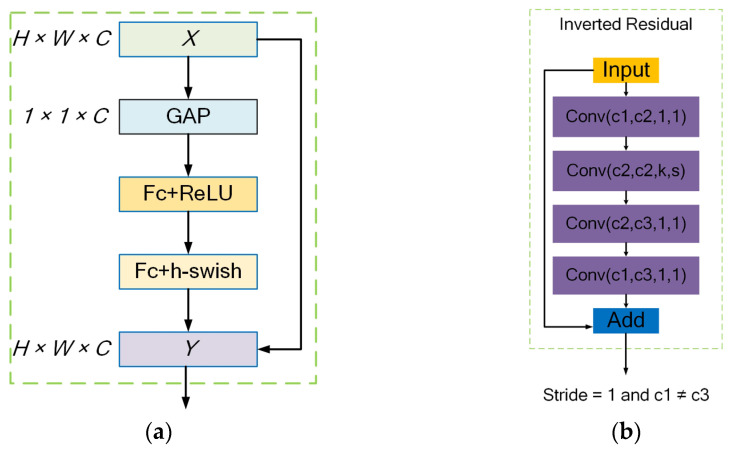
(**a**) Structure of Squeeze-and-Excitation module; (**b**) inverted residual structure.

**Figure 8 sensors-23-06738-f008:**
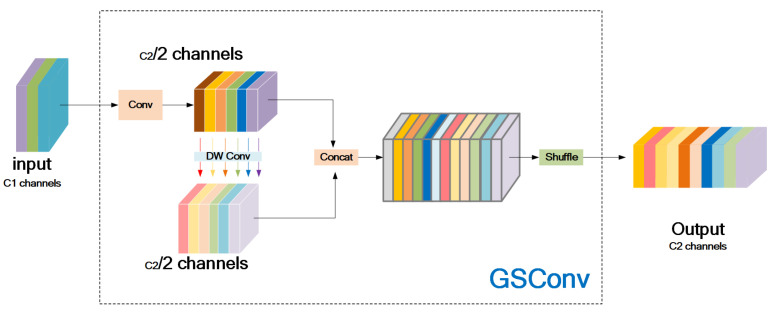
The structure of the GSConv module.

**Figure 9 sensors-23-06738-f009:**
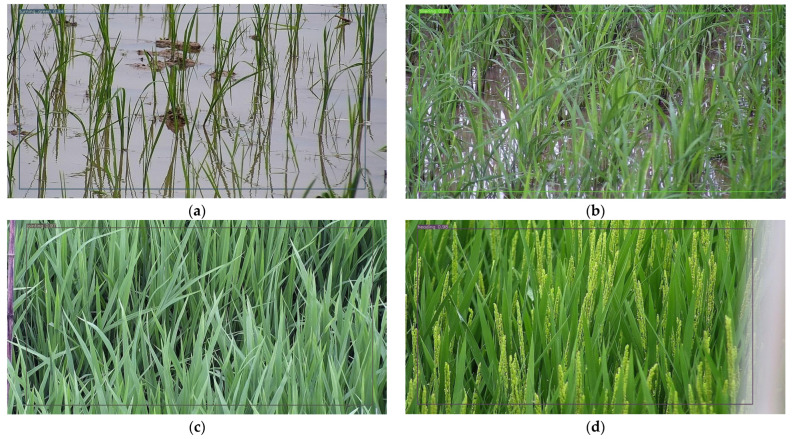
Recognition results of rice growth period. (**a**) Turning green growth period; (**b**) tillering growth period; (**c**) jointing growth period; (**d**) heading growth period; (**e**) milky growth period.

**Figure 10 sensors-23-06738-f010:**
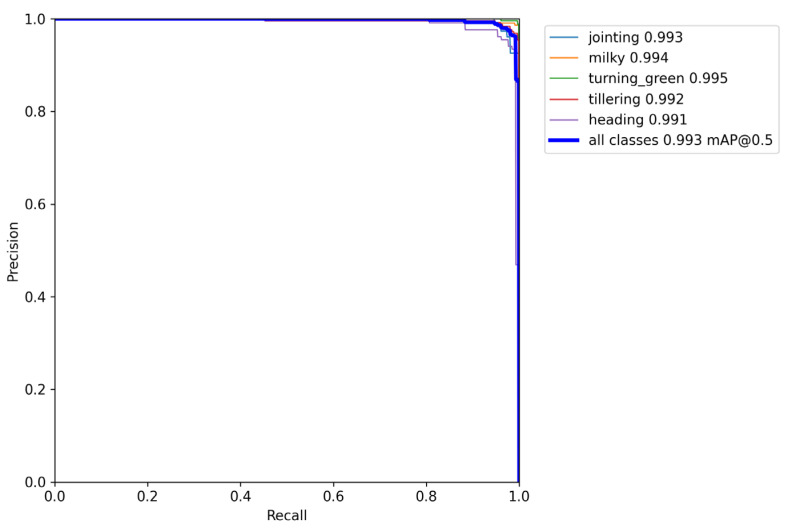
*P*–*R* curves of the proposed model.

**Table 1 sensors-23-06738-t001:** Description of rice growth period dataset.

Growth Period	Training	Validation
turning green	548	137
tillering	912	228
jointing	834	208
heading	592	148
milky	990	247

**Table 2 sensors-23-06738-t002:** Comparison of the evaluation indexes of the combination of different modules.

Model	Params (MB)	GFLOPs	*P* (%)	*R* (%)	*mAP* (0.5) (%)	*mAP* (0.5:0.95) (%)
YOLOv5s(Baseline)	7.06	16.3	98.5	98.9	99.4	97.3
YOLOv5s + MobileNetV3	1.39	2.5	98.2	97.4	99.3	89.4
YOLOv5s + GsConv	6.58	15.2	92.6	98.0	97.3	73.4
YOLOv5s + MobileNetV3 + GsConv	1.24	2.3	96.2	98.9	98.7	94.2

**Table 3 sensors-23-06738-t003:** Comparison of evaluated indicators of different models.

Model	Params (MB)	GFLOPs	*P* (%)	*R* (%)	*mAP* (0.5) (%)	*mAP* (0.5:0.95) (%)
Faster R-CNN	13.7	37.0	98.1	96.3	96.7	93.2
YOLOv4	10.6	18.2	93.5	92.6	90.5	88.7
YOLOv7	36.5	103.2	98.7	97.4	99.6	98.6
YOLOv8	11.1	28.4	99.1	99.0	99.5	98.8
Small-YOLOv5	1.24	2.3	96.2	98.9	98.7	94.2

**Table 4 sensors-23-06738-t004:** Comparison results of different lightweight detection models.

Model	Params (MB)	GFLOPs	*P* (%)	*R* (%)	*mAP* (0.5) (%)	*mAP* (0.5:0.95) (%)
YOLOv4-tiny	4.6	5.2	92.1	90.7	88.9	86.7
YOLOv7-tiny	5.1	8.2	93.2	95.1	95.6	93.7
YOLOv5n	1.76	4.1	94.6	95.1	98.8	91.7
YOLOv8n	3.0	8.1	99.3	99.3	99.5	98.8
Small-YOLOv5	1.24	2.3	96.2	98.9	98.7	94.2

## Data Availability

Not applicable.

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
