# Peer review of "A Lightweight Recognition Method for Rice Growth Period Based on Improved YOLOv5s"

_sensors, 2023, doi:10.3390/s23156738_

Round 1

Reviewer 1 Report

The objective of this paper is to introduce a lightweight model for accurately recognizing the growth period of rice. The proposed method is validated through experiments, demonstrating its ability to achieve comparable results to YOLOv5s on their own dataset while utilizing fewer parameters and GFLOPS.

However, the contributions of the paper are limited and lack innovation, primarily focusing on two key aspects: 1) replacing the backbone of YOLOv5s with MobileNetv3, and 2) utilizing GsConv, a novel convolutional layer proposed by Hulin Li, as a replacement for traditional convolution layers.

The overall quality of English in the manuscript is acceptable, and is clear to read and understand.

Reviewer 2 Report

This paper proposes a lightweight model, Small-YOLOv5s, which uses deep learning to recognize network models. Small-YOLOv5s chooses MobileNetV3 to replace the backbone network of YOLOv5s, and introduces a lighter convolution method GsConv to reduce model complexity while maintaining accuracy. This model can help growers time field operations correctly and improve field management efficiency. This article has several shortcomings:

1. Whether the dataset of the method is collected from a single farmland can be validated and tested in a diverse farmland environment.

2. Can you explain in more detail the details of the comparison between the experimental results and other models?

3. Insufficient description details for the dataset

4. In the face of the unfavorable growth period of rice, the description of the experimental results of the model is not specific enough.

This paper proposes a lightweight model, Small-YOLOv5s, which uses deep learning to recognize network models. Small-YOLOv5s chooses MobileNetV3 to replace the backbone network of YOLOv5s, and introduces a lighter convolution method GsConv to reduce model complexity while maintaining accuracy. This model can help growers time field operations correctly and improve field management efficiency. This article has several shortcomings:

1. Whether the dataset of the method is collected from a single farmland can be validated and tested in a diverse farmland environment.

2. Can you explain in more detail the details of the comparison between the experimental results and other models?

3. Insufficient description details for the dataset

4. In the face of the unfavorable growth period of rice, the description of the experimental results of the model is not specific enough.

Reviewer 3 Report

Please find the comments referring to the paper as an attachment.

Round 2

Reviewer 1 Report

Q1:The experiment did not include a comparison with the current state-of-the-art (SOTA) techniques, but instead conducted internal ablation experiments. It is advisable to perform a comparative analysis with existing approaches for rice classification or other relevant advanced algorithms. By conducting a vertical comparison, the experiment's results can be better contextualized and evaluated. This would enhance the robustness and significance of the findings.(The content of Table 5 and Table 4 is also the ablation of its own algorithm).

Q2: The growth stage classification proposed by the author holds some significance for agricultural production and rice cultivation management. Different growth stages of rice have distinct requirements concerning light, temperature, water, and nutrients. Thus, comprehending and identifying the growth cycle of rice can optimize its growth and yield potential. However, a simple categorization of growth stages might not offer substantial practical value for actual farmers. Therefore, it is suggested that the author considers incorporating parameter indicators such as root growth, leaf color, stem height, leaf area, and tiller count for measurement and monitoring purposes. By quantitatively assessing these indicators, a more accurate evaluation and determination of rice's growth status and developmental stages can be achieved. Instead of relying solely on a rudimentary growth stage classification, the author can include these indicators in the experiment for more precise assessment(recommend due consideration).

Reviewer 3 Report

All comments of the reviewer have been included in the revised version of the paper. I recommend publication of this paper in its current form.

Author Response

Thank you very much for your advice and approval!

Round 3

Reviewer 1 Report

Looking forward to seeing the future author's multi-feature fusion rice growth stage identification method and related rice phenotype information datasets compiled for public use.